# Riding the Plumes: Characterizing Bubble Scavenging Conditions for the Enrichment of the Sea-Surface Microlayer by Transparent Exopolymer Particles

**Tiera-Brandy Robinson [1],[*]** , **Helge-Ansgar Giebel [2]** and **Oliver Wurl [1]**

1 Institute for Chemistry and Biology of the Marine Environment, University of Oldenburg, 26382-Wilhelmshaven, Germany
2 Institute for Chemistry and Biology of the Marine Environment, University of Oldenburg, 26111-Oldenburg, Germany
* Correspondence: tiera-brandy.robinson@uol.de

**Abstract:** Transparent exopolymer particles (TEP) act as a major transport mechanism for organic matter (OM) to the sea surface microlayer (SML) via bubble scavenging, and into the atmosphere via bubble bursting. However; little is known about the effects of bubble scavenging on TEP enrichment in the SML. This study examined the effects of several bubbling conditions and algae species on the enrichment of TEP in the SML. TEP enrichment in the SML was enhanced by bubbling, with a larger impact from bubbling rate than bubble size and increasing enrichment over time. Depth profiles showed that any TEP aggregates formed in the underlying water (ULW) were rapidly (<2 min) transported to the SML, and that TEP was entrained in the SML by bubbling. Species experiments determined that the presence of different phytoplankton species and their subsequent release of precursor material further enhance the effectiveness of TEP enrichment via bubble scavenging.

**Keywords:** bubble scavenging; sea surface microlayer; transparent exopolymer particles

## 1. Introduction

The sea surface microlayer (SML) is a thin layer at the top of the ocean with its own distinct chemical, biological and physical characteristics [1–3]. This layer, which is 10–1000 μm thick, acts as the boundary layer between the ocean and atmosphere. All matter transferring between the ocean and atmosphere must pass through the SML. In addition, the SML is a primary source of organic material (OM) transferred as marine aerosols to the atmosphere [4,5]. The SML is a gelatinous film often enriched by many biological and chemical parameters in comparison to the underlying water (ULW), including phytoplankton, bacteria and surfactants [6–9]. It is the gelatinous nature which allows the visual observation of highly enriched areas—termed "slicks"—under low wind speeds below 5 m s$^{-1}$ due to the dampening of capillary waves [3]. The gel component of the SML is caused by the creation and accumulation of extracellular polymeric substances (EPS), the largest faction of which is transparent exopolymer particles (TEP) [3].

TEP are a class of non-living organic material made of acidic polysaccharides which are stainable by Alcian blue [10]. They are operationally defined as gel particles retained on membrane filters with pore sizes of 0.2 to 0.4 μm. TEPs are formed from the self-adherence of colloidal precursor materials which are "sticky" by nature and will adhere to themselves and to other forms of matter, making them a key component for OM transfer. When attached to dense matter, these aggregates will sink down as "marine snow", and when positively buoyant, which is their natural state, will rise up to the SML.

Phytoplankton have been found to be the primary source of dissolved precursor material for the formation of TEP. Peaks in the formation of TEP typically correspond to phytoplankton blooms [11–13]. Phytoplankton are known to release precursor material by active or passive permeation during the growth and stationary phases, and cell lysis during senescence [14]. Furthermore, lack of nutrients or physical stressing of phytoplankton species has been shown to increase the release of precursor material and TEP [14–16]. While diatoms have been shown to be the group of phytoplankton to produce the largest amounts of precursor material, some species of dinoflagellates and other non-diatom species have also been found to produce comparable concentrations [14]. Additionally, bacteria can also produce TEP [14,17] and are colonizers of TEP aggregates [18–20]. For example, Mari and Kiorboe [21] found that up to 68% of total bacterial cells were attached to TEP. Thus, TEP play a large role in both the microbial loop and biological carbon pump.

The accumulation and enrichment of TEP and attached OM in the SML have been found to be greatly enhanced by bubble-associated scavenging [22,23]. Enrichment of TEP in the SML can alter the sea surface hydrodynamics and act as a physicochemical barrier, suppressing the air–sea exchange of carbon dioxide and other trace gasses [1,3,24]. Additionally, because of bubble bursting, TEP, with its attached OM and other marine particles, can be released into the atmosphere [25–28]. These organic-rich aerosols can potentially contribute to cloud-condensing nuclei and ice-nucleating particles [29–31].

However, most previous SML-related studies have focused on aerosol composition [25–31] and not specifically on TEP abundance, even though TEP is typically enriched in the SML. In addition, the aforementioned studies have focused on the effects of bubble bursting rather than bubble scavenging, i.e., the transport of OM, including TEP, on rising bubble plumes. For this reason, most studies which involve bubbling are more concerned with resulting properties of sea spray particles than the bubbling mechanisms themselves. Thus, there is a need to consider the effects of bubble scavenging on TEP enrichment in the SML, including bubble size, bubbling rate and time, and TEP enrichment in the SML.

The purpose of the present study was to investigate the effect of varying bubbling conditions on the effectiveness of TEP enrichment. We present results from four experiments; three which focus on bubbling time, size, concentration and vertical movement via abiotic processes. In the fourth experiment, natural seawater was spiked with different species of phytoplankton to investigate the effect of biotic processes on TEP enrichment. A mechanistic understanding on how TEP becomes enriched in the SML is crucial to understand the formation of organic-rich aerosols via sea spray generation.

## 2. Methods & Materials

### 2.1. Tower Schematic

In order to measure TEP abundance in both ULW and SML, triplicate bubble towers were built from acrylic glass (Figure 1). The design was based on the one used by Zhou, Mopper and Passow [22], but was enlarged and adjusted so that the SML could be sampled. The towers measured 2 m high with a column (165 cm tall, 25 cm in diameter) below and a box ($35 \times 50 \times 50$ cm$^3$) above. The tower held a total volume of 186 L. However, during all experiments, the upper box was filled halfway to allow space for SML sampling, thus a total volume of 142 L was used. Glass frits (25 mm diameter) of varying porosity were used to produce bubbles of sizes between 55 and 2400 µm. Porosity grades of 1, 3 and 4 (European Standard) were used, representing ranges of pore sizes of 100–160 µm, 16–40 µm and 10–16 µm, respectively, and produced average bubble diameters of 1.35 µm, 0.68 µm and 0.58 µm, respectively, which were within the natural size spectrums of bubble plumes created by wave breaking [32,33]. At the bottom of the tower, a valve and detachable chamber for the frits was constructed. It allowed an easy and quick change of fritz with different porosity, for example, changing to different bubble sizes using the same bulk water. In order to change the bubbling rate, each tower was supplied with an individual runoff line from the main source of compressed air. Each individual runoff line could be adjusted, and the air flow rate was measured. Taps were built at the

following depths: 5, 50, 100 and 150 cm in the tower column to allow discrete sampling of water, which equated to depths of 40, 85, 135 and 185 cm below the water surface, respectively. SML samples were collected from above, using the glass plate technique [34,35]. In order to measure the bubble size spectrum, images of bubbles were taken using a video camera (GigE Prosilica GT, Allied Vision) after each experiment, so that imaging would not affect bubbling time. Images were then analyzed using an Image J software protocol similar for TEP image analysis, suggested by Engel et al. [36]. Clear millimeter paper was placed in the tower and imaged in addition, being done to account for the field of view on camera and known size in Image J.

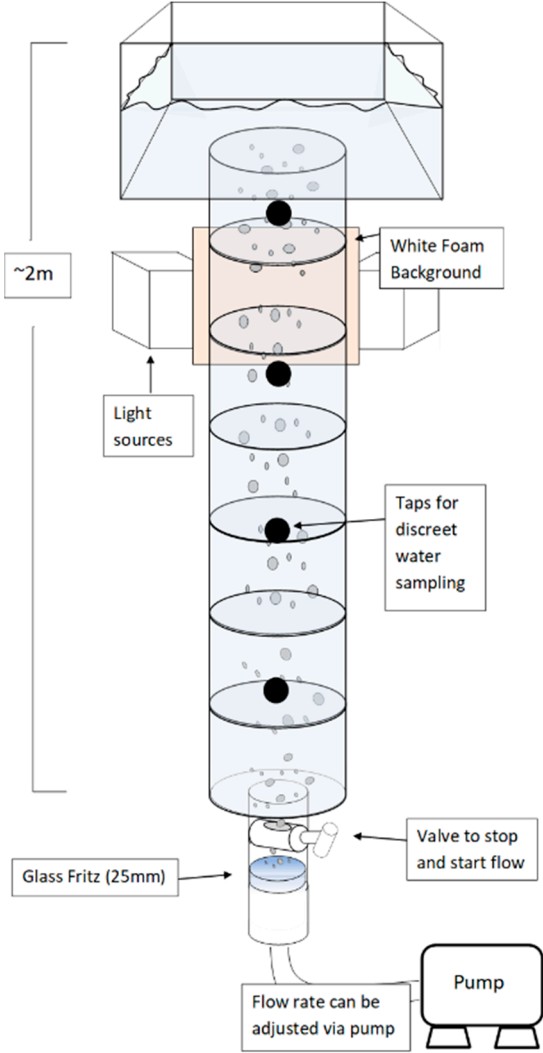

**Figure 1.** Schematic view of one tower (towers were built in triplicate) with detachable glass frits, adjustable air flow rate, discrete water sampling taps and surface microlayer (SML) sampling box.

*2.2. Experimental Setup*

2.2.1. Overall Setup

Natural seawater was collected in April 2019, from the Jade Bay in Wilhelmshaven, Germany, at the beginning of the experiments. The seawater was stored in concrete basins at the Institute for Chemistry and Biology of the Marine Environment (ICBM) for up to 10 days. Fresh seawater was collected for experiments 3–4. Prior to each experiment, 500 L of water was pumped (Eden 135, Conrad, Germany) into an aquarium and then circulated for 2–4 h through a fine fleece filter (Rollermat, Thielling GmbH, Melle, Germany) with a density of 60 g/m$^2$. The fleece-filtered water was pumped (Eden 135, Conrad,

Germany) in each tower, and experiments were subsequently run. For experiments 1, 2 and 4, replicate towers were used to alter experimental parameters (i.e., flow rate, bubble size or phytoplankton species) and the experiment with all towers was repeated in triplicate. For experiment 3, which required only one tower per replicate, all three towers were instead run simultaneously to get the required triplicate. After each experiment, all towers were rinsed with 10%-HCl, milli Q water and finally fleece-filtered seawater to eliminate any residual contamination from previous experiments.

### 2.2.2. Experiment 1—Bubbling Time and Production Rate

Glass frits with the smallest porosity (porosity 4) were used, as their theoretical bubble size production most closely mimics those produced by natural wave breaking [37,38]. However, the flow rate for each tower was changed to produce three different intensities of bubble plumes. While the concentration of bubbles could not be analyzed from images, we use the following equation (Equation (1)) to estimate the bubble production rate. The production rate was calculated from the air flow rate F (mL/min) and the bubble volume V (mm$^3$) using the averaged diameter of the bubbles derived from the images. Bubble size was given as diameter length (mm) for comparison with previous literature.

$$bubbles/min = F \times (V/1000) . \tag{1}$$

Assuming bubbles rose in about 30 s to the surface of our towers (volume 142 L), we used the bubble production rate (Equation (1)) to estimate a bubble density ranging from $3.1 \times 10^7$ to $5.6 \times 10^8$ bubbles m$^{-3}$. The estimated bubble density was slightly larger than natural densities in a range of $10^5$ to $10^7$ [39,40], but the common acoustic techniques to count natural bubbles is limited to small-sized bubbles, for example, up to 130 μm [39], and, therefore, may underestimate bubble densities. Bubbling was run continuously for 60 min with sampling at 0, 5, 10, 15, 30 and 60 min, as well as 30 and 60 min after bubbling was stopped (referred to as 90 and 120 min). ULW water was sampled from a tap at 135 cm depth below the surface.

### 2.2.3. Experiment 2—Bubble Size

Glass frits with three different porosities (porosity 4, 3 and 1) were used to produce three different bubble sizes. The first two were chosen, as they were those which most closely resembled the size range with the highest density produced by a breaking wave [37,38] and which occur below the Hinze scale. The Hinze scale determines the power–law relation of bubble creation within breaking waves and states that the bubble size distribution for bubbles smaller than 1 mm is determined by entrainment, while bubbles larger than 1 mm have a size distribution dependent on turbulent fragmentation [41]. The third bubble size in our experiment (porosity 1) was chosen to investigate how larger bubbles (1.35 ± 0.34 mm), which naturally occur in wave breaking and which reside above the Hinze scale, can affect TEP transport. Instead of continuous bubbling, which was used for experiment 1, bubbling times of 2 min and 10 min were used. Initial SML and ULW concentrations were measured in samples taken each time before bubbling was started. Thus, the initial samples were taken, bubbling was run for 2 min, bubbling stopped and samples were taken. Then, another set of initial samples were taken using the same bulk water, bubbling was run for 10 min, stopped and samples were taken.

### 2.2.4. Experiment 3—Depth Profiles of TEP

To test if TEP was being produced and transported from the ULW or produced in the SML, an experiment was run with the size and abundance of bubbles which allowed maximum transport, determined from the previous experiments (porosity 4, 16.5 L/min). For this experiment, samples were taken from four depths in the ULW: 45, 85, 135, 185 cm and the SML to show depth profiles in the towers. Because there was no change in bubble type, each tower was used as a replicate. A control experiment was run first, which was done to see the natural vertical movement of OM in the tower

without bubbling. Then, a bubbling experiment was run, the same as experiment 2, with a bubbling time of 2 and 10 min.

### 2.2.5. Experiment 4—Effect of Phytoplankton Species on TEP Enrichment from Bubble Scavenging

Unlike the previous experiments, the objective for experiment 4 was to investigate at the biotic formation and transport of TEP via bubble scavenging, and thus unfiltered seawater was used for both the control and experiments. Seawater was pumped into the aquarium and then into the towers in order to maintain similar protocol methods for inter-comparison of experimental results. Three species of phytoplankton were grown in triplicate cultures of 450 mL (*Thalassiosira rotula* and *Lingulodinium polyedra*) or 200 mL (*Phaeocystis* spp.) in a control growth chamber under 12:12 light dark cycles with 90 μmol photon m$^{-2}$s$^{-1}$ at 18 °C for two weeks prior to the experiment. All species chosen are known to occur in North Sea water and thus would not have adverse effects to being added to natural seawater from the North Sea. *T. rotula* was chosen as a known TEP-producing diatom [13]. *L. polyedra* was chosen as a dinoflagellate whose production of TEP is unknown, but which has large effects as the main source of "red tide" events in coastal waters. A *Phaeocystis* species was also included because of their capability of forming floating colonies with highly surface-active properties. In order to ensure equal cell concentration, triplicate cultures were mixed and subdivided just prior to the first experimental replicate. However, because all experiments could not be run on the same day, cell counts for each culture were determined via microscopy before every experimental replicate. The towers were filled with approximately 107 L of unfiltered seawater and spiked with 450 mL or 200 mL of phytoplankton culture 24 h prior to the experiment to allow for their adjustment. The final 35 L of seawater was added the next day, prior to the experiment, so that the mixing of water would resuspend any phytoplankton on the bottom or walls of the towers. Because of the limited number of towers (three) versus species plus control (four) a hot bunking system was used, i.e., the first replicate experiment was run with *T. rotula*, *Phaeocystis* spp. and a control, then *T. rotula*, *L. polyedra* and a control, and so on. With this approach, all possible combinations of species, plus a control every time, were run in the three towers. Glass frit porosity, flow rate, bubbling time and sampling protocols were all kept the same as for experiment 3, with the exception that ULW was only sampled at 135 cm depth.

### 2.3. Measurements

### 2.3.1. Transparent Exopolymer Particles (TEP)

TEP were analyzed spectrophotometrically [10]. Discrete samples were filtered onto 0.2 μm polycarbonate filters under low vacuum (<100 mmHg) and stained with an Alcian blue solution (0.02 g Alcian blue in 100 mL of acetic acid solution, pH 2.5) for 5 s. We used a filter membrane with a pore size of 0.2 μm in order to collect both large and smaller colloidal TEP material. Samples were stored at −18 °C until analyzed. The stained filters were placed in glass vials with 5 mL extraction solution (80% $H_2SO_4$). The filters were allowed to extract for 2 h on a shaker with gentle agitation to reduce bubble formation. A xanthan gum (Carl Roth) standard was used to calibrate Alcian blue stock solution according to Passow and Alldredge [10]. For this reason, TEP concentrations are shown in relation to xanthan gum equivalence (μg XG eq/L). Recent calibration issues with xanthan gum were not observed in our studies, and thus the new method by Bittar, et al. [42] was not required.

### 2.3.2. Chlorophyll *a* (Chl *a*)

Due to the limited volume of SML in the upper box of our towers and sampling time constraints, in vivo Chl *a* was measured instead of discrete Chl *a* samples. In vivo Chl *a* was measured with a hand fluorometer (Turner Designs, AquaFluor) and related to μg of Chl *a* using a calibration factor between filtered Chl *a* (Chl *a* standard in ethanol as reference) and in vivo absorbance.

### 2.3.3. Microbial Counts

The protocol from Giebel, et al. [43] was used for the determination of heterotrophic prokaryote and cyanobacteria abundance. Water samples were fixed with glutaraldehyde (1% final concentration), incubated at room temperature for 1 h and stored at −18 °C until further analysis. Cells were stained with SYBR™ Green I (2.5 mM final concentration, Molecular Probes, Schwerte, Germany) for 30 min in the dark. Samples were measured on a flow cytometer (C6 Flow Cytometer, BD Bioscience), and cells were counted according to side-scattered light and emitted green and red fluorescence. We used 1.0 µm beads (Fluoresbrite Multifluorescent, Polysciences Europe, Hirschberg, Germany) as an internal reference to monitor the performance of the device.

### 2.3.4. Salinity/Temperature

Salinity, temperature and pH of the seawater was measured in the aquarium prior to filling the towers using a refractometer (RHS-10ATC) and multiparameter PCD (PCD650, EuTech Instruments, Singapore) at the beginning of each experiment/replicate to monitor for any significant changes.

### *2.4. Statistical Analyses*

Statistical analyses of the data set were performed using Graphpad PRISM Version 8. Differences, null hypothesis testing and correlation were considered significant when $p < 0.05$. Greenhouse–Giesser corrections were used for analysis of variance (ANOVA) testing in experiment 1. Further post hoc Tukey analysis was run for comparison of means when the ANOVA test yielded significant differences. Unless otherwise indicated, results are presented as means ± standard deviation. Enrichment factors (EF) were calculated as the ratio of concentrations in the SML sample to that of the corresponding subsurface bulk water sample.

## 3. Results

### *3.1. Experiment 1—Effect of Bubble Concentration and Time on TEP*

Because of stable ULW concentrations, TEP enrichment closely matches their concentration trends in the SML. Figure 2A shows that TEP enrichment caused by bubbling was lowest in the tower with the lowest flow rate (14.5 L/min). In this case, the enrichment factor (EF) increased from 0.9 ± 0.1 to 2.0 ± 0.2. Similarly, the highest increase in EF from 1.3 ± 0.2 to 6.7 ± 1.2 was observed, with the highest flow rate (16.5 L/min) used for bubbling. Post hoc Tukey's analysis showed that increases in EF were significantly higher using a flowrate of 16.5 L/min for bubbling ($p < 0.001$, $n = 24$), but insignificantly different between flowrates of 14.5 and 15.5 L/min ($p = 0.07$, $n = 24$). Prokaryote cell concentrations in the ULW were relatively constant (largest decrease of $1.24 \times 10^6$ cells/mL in 15.5 L/min tower) during and after bubbling, so that the EF closely matched SML trends. Prokaryote enrichment in the SML after 60 min bubbling (0 min to 60 min) increased by 13%, 36% and 200% for 14.5, 15.5 and 16.5 L/min, respectively. Additionally, decreases were observed after bubbling stopped (60 min to 120 min) from the 15.5 L/min and 16.5 L/min flow rates (2% and 45% decrease). Enrichment factors for 16.5 L/min ranged from 1.1 to 3.4 and were significantly higher (post hoc Tukey, $p < 0.001$, $n = 8$), while the 15.5 and 14.5 L/min flow rates had EF ranges of 1.0 to 1.2 and 1.0 to 1.4, respectively, with no significant difference ($p = 0.86$, $n = 8$). Increases in enrichment were highest for cyanobacteria (Figure 2C) compared to TEP (Figure 2A) and prokaryotes (Figure 2B). Enrichment factors for cyanobacteria increased from 1.9 to 3.1 and 1.7 to 17.6 for 15.5 and 16.5 L/min, respectively, with a maximum EF of 17.6 after 60 min of bubbling at 16.5 L/min flow rate. Post hoc Tukey's analysis showed that increases in EF for cyanobacteria were significantly higher in 16.5 L/min bubbling ($p < 0.001$, $n = 8$), but not significantly different between 14.5 and 15.5 L/min ($p = 0.90$, $n = 8$).

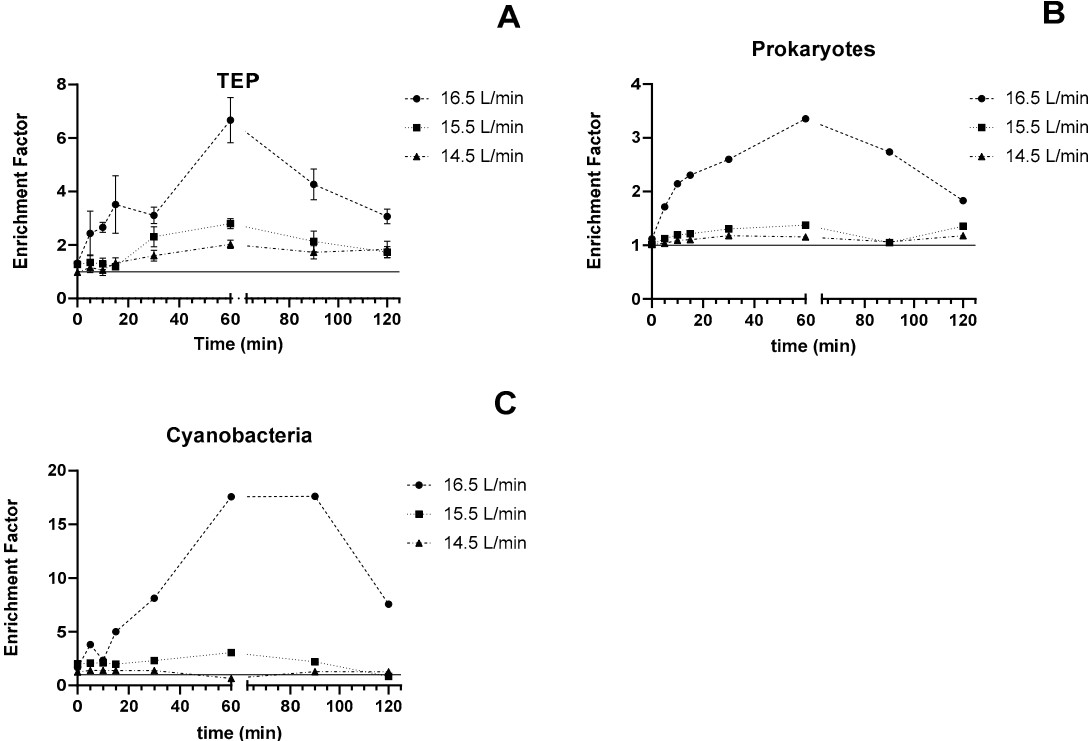

**Figure 2.** Temporal enrichment (enriched when enrichment factor (EF) > 1) of transparent exopolymer particles (TEP) (**A**), prokaryotes (**B**) and cyanobacteria (**C**) in three experiments with flow rates of 14.5, 15.5 and 16.5 L/min. Results are given as SML enrichment factors (i.e., concentration in SML/concentration in underlying water (ULW)). Bubbling was stopped at 60 min, noted by the break in the *x*-axis.

## 3.2. Experiment 2—Effect of Bubble Size on TEP

The average diameter of bubbles were 0.58 ± 0.06 mm (porosity 4), 0.68 ± 0.1 mm (porosity 3) and 1.35 ± 0.38 mm (porosity 1) for each tower. The flow rate was set to 15.5 L/min in all towers to create an equal total volume of bubbles. This means that the tower characterized with a bubble diameter of 0.58 mm had equivalently higher numbers of bubbles ($3.02 \times 10^8$ after 2 min and $1.51 \times 10^9$ after 10 min) compared with the tower characterized with a bubble diameter of 1.35 mm bubbles, yielding lower numbers ($1.90 \times 10^7$ after 2 min and $9.52 \times 10^7$ after 10 min). Bubbling with large bubbles (1.35 ± 0.38 mm) had the greatest increase of TEP in the SML after 10 min, with an increase of 3574 µg XG eq/L (52% increase), and the greatest EF increase from 1.1 to 1.36. However, this increase was insignificantly higher compared to each of the other bubble sizes (Table 1). Likewise, considering A/B bubbling enrichment (i.e., EF after bubbling/ EF before bubbling), all parameters showed no significant effect from bubble size (all $p > 0.1$). Similar trends between bubble sizes for all parameters are shown in Figure 3.

**Table 1.** Summary table showing enrichment factor (EF) from after bubbling/before bubbling (A/B) ± standard deviation (SD) for TEP from all experiments, as well as statistical analysis of enrichment of TEP during experiments. Significant results are in bold. Experiment 3 shows changes in TEP concentrations after 2 and 10 min (ΔTEP), results were analyzed as depth profiles, and thus no statistics were used.

| **Experiment 1** | | | | | | |
|---|---|---|---|---|---|---|
| | EF (A/B 60 min bubbling) ± SD | Test | Parameter (bubbling flow rate) | *n* | mean diff ± SE | *p*-value |
| | | 2-way ANOVA | **16.5 L/min, 15.5 L/min, 14.5 L/min** | **9** | | **0.0361** |
| 16.5 L/min | 5.03 ± 0.33 | | **16.5 L/min vs. 15.5 L/min** | **24** | **1.619 ± 0.30** | **<0.0001** |
| 15.5 L/min | 2.21 ± 0.27 | Post-hoc Tukey | **16.5 L/min vs. 14.5 L/min** | **24** | **1.915 ± 0.30** | **<0.0001** |
| 14.5 L/min | 2.12 ± 0.42 | | 15.5 L/min vs. 14.5 L/min | 24 | 0.296 ± 0.13 | 0.0744 |
| **Experiment 2** | | | | | | |
| | EF(A/B) ± SD | | Parameter (bubbling time–bubble size) | *n* | mean diff ± SE | *p*-value |
| 0.58 mm−2 min | 0.97 ± 0.07 | | 2 min–0.58 mm vs. 0.68 mm | 6 | −0.55 ± 0.27 | 0.1504 |
| 0.68 mm−2 min | 1.52 ± 0.50 | | 2 min–0.58 mm vs. 1.35 mm | 6 | −0.43 ± 0.27 | 0.2979 |
| 1.35 mm−2 min | 1.39 ± 0.36 | Post-hoc Tukey | 2 min–0.68 mm vs. 1.35 mm | 6 | 1.23 ± 0.27 | 0.8938 |
| 0.58 mm−10 min | 0.99 ± 0.08 | | 10 min–0.58 mm vs. 0.68 mm | 6 | −0.18 ± 0.27 | 0.7828 |
| 0.68 mm−10 min | 1.18 ± 0.14 | | 10 min–0.58 mm vs. 1.35 mm | 6 | −0.33 ± 0.27 | 0.4559 |
| 1.35 mm−10 min | 1.33 ± 0.20 | | 10 min–0.68 mm vs. 1.35 mm | 6 | −0.15 ± 0.27 | 0.8423 |

| **Experiment 3** | | | | | |
|---|---|---|---|---|---|
| | Without Bubbling | | With Bubbling | | |
| Depth (cm) | ΔTEP (µg XG eq L$^{-1}$) 2 min | ΔTEP (µg XG eq L$^{-1}$) 10 min | ΔTEP (µg XG eq L$^{-1}$) 2 min | ΔTEP (µg XG eq L$^{-1}$) 10 min | |
| 0 | −125 ± 151 | −122 ± 134 | 44 ± 481 | 1941 ± 1344 | |
| 40 | −113 ± 357 | 4 ± 124 | −235 ± 70 | 121 ± 88 | Statistical tests not applied |
| 85 | 199 ± 309 | 87 ± 55 | −61 ± 128 | 114 ± 157 | |
| 135 | 75 ± 190 | 221 ± 325 | 118 ± 107 | 46 ± 62 | |
| 185 | 70 ± 79 | 102 ± 185 | −4 ± 94 | −171 ± 119 | |

| **Experiment 4** | | | | | | |
|---|---|---|---|---|---|---|
| | EF(A/B) ± SD | Test | Parameter (bubbling time–species) | *n* | mean diff ± SE | *p*-value |
| 2 min—Control | 1.19 ± 0.10 | | 2 min—Control vs. *T. rotula* | 6 | −0.5787 ± 0.48 | 0.7469 |
| 2 min—*T. rotula* | 1.77 ± 0.49 | | 2 min—Control vs. *Pheaocistus* spp. | 6 | −0.7784 ± 0.48 | 0.5591 |
| 2 min—*Phaeocystis* spp. | 1.97 ± 0.72 | Post-hoc Tukey | 2 min—Control vs. *L. polydra* | 6 | 0.2523 ± 0.48 | 0.9684 |
| 2 min—*L. polyedra* | 0.94 ± 0.10 | | **10 min—Control vs. *T. rotula*** | **6** | **−1.869 ± 0.48** | **0.0424** |
| 10 min—Control | 1.25 ± 0.33 | | 10 min—Control vs. *Pheaocistus* spp. | 6 | −0.9975 ± 0.48 | 0.3726 |
| 10 min—*T. rotula* | 3.12 ± 1.27 | | **10 min—Control vs. *L. polydra*** | **6** | **−2.276 ± 0.48** | **0.0131** |
| 10 min—*Phaeocystis* spp. | 2.25 ± 0.28 | | | | | |
| 10 min—*L. polyedra* | 3.52 ± 1.15 | | | | | |

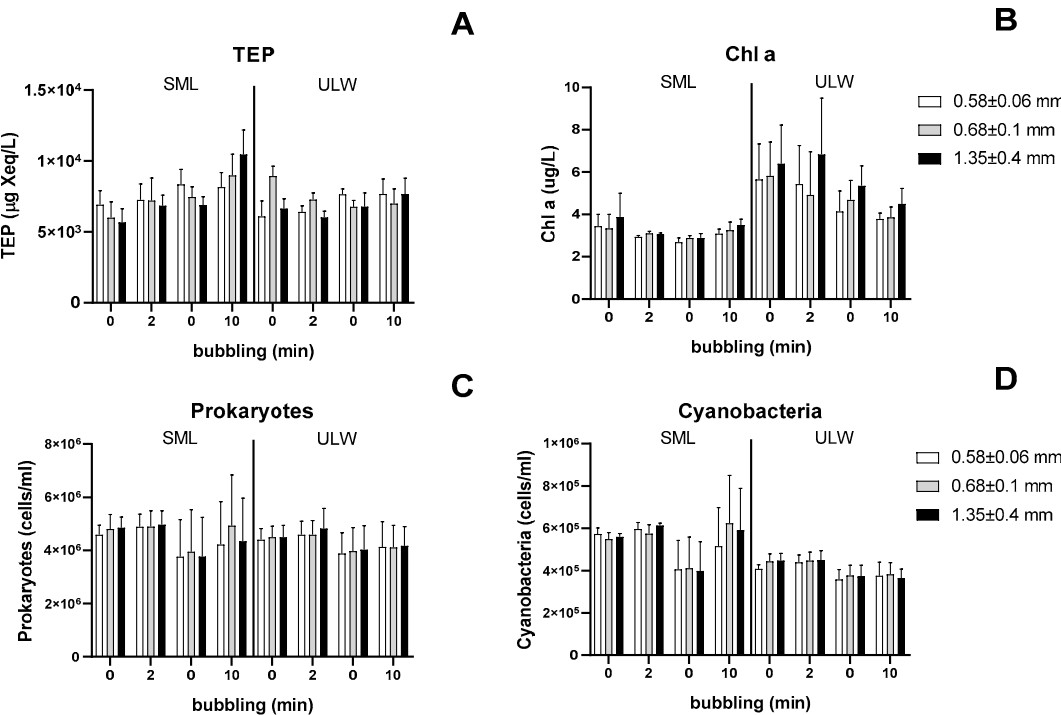

**Figure 3.** Total concentrations from initial sampling (0), after 2 min bubbling (2), initial sampling again (0), after 10 min (10) bubbling for TEP (**A**), Chlorophyll *a* (**B**), heterotrophic prokaryotes (**C**) and cyanobacteria (**D**). Results are grouped into SML samples (left) and ULW samples (right). Legend shows the average ± SD diameter of bubbles.

### 3.3. Experiment 3—Effect of Bubbling on Depth Profiles of TEP and Chl a

Background enrichment (enrichment over time without bubbling) and bubbling experiments were conducted to investigate the depth profiles of TEP and whether vertical fluxes of TEP could be observed. Background enrichment experiments (Figure 4A,C) showed that no changes in the enrichment of the SML occurred for either Chl *a* or TEP within 2 and 10 min. In comparison, during bubbling (Figure 4B,D), SML enrichment was observed for both 2 and 10 min for TEP (EF = 1.9 ± 0.1, EF = 3.5 ± 1.2) and Chl *a* (EF = 1.3 ± 0.1, EF = 1.3 ± 0.1). Chl *a* abundance had a higher increase in both the SML and ULW after 10 min compared to 2 min, resulting in equal EF. There was both rapid transport of existing TEP to the SML and creation of new TEP, either within the SML or in the ULW which was immediately transported to the SML as outlined in the following section. It is important to note that background and bubbling enrichment experiments were conducted using the same water, i.e., water had been inside the towers for approximately 20 min when initial samples were taken for bubbling experiments. This shows that, while in stagnant seawater, SML enrichment does not occur after 10 min, yet it does after 20 min by comparing Figure 4A,B. Because of the differences between total volume of SML and ULW, changes in ULW concentrations did not equate to equal changes in SML concentrations, and thus changes in ULW must be considered separately from the SML. Decreasing TEP concentrations in three of four ULW depths after 2 min and increasing in three of four ULW depths after 10 min suggested that pre-existing TEP in the ULW was rapidly (<2 min) brought to the surface and new TEP aggregates were formed in the ULW between 2 to 10 min.

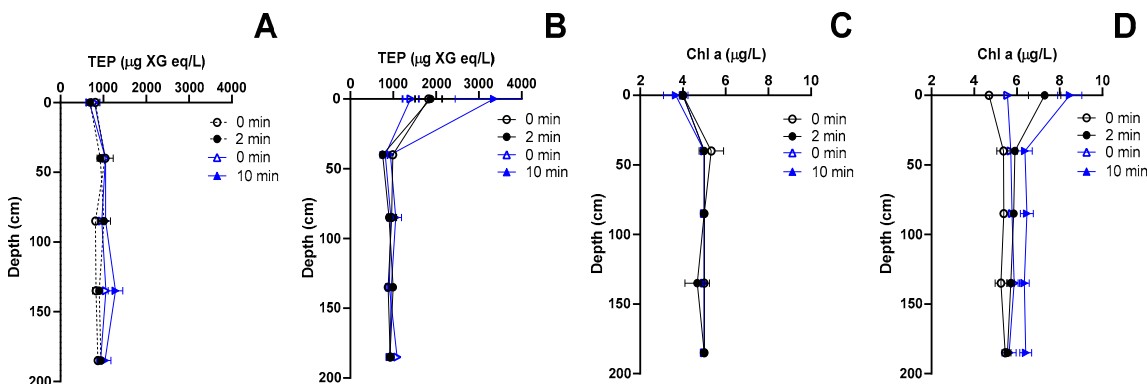

**Figure 4.** Depth profiles for TEP (**A**,**B**) and Chl *a* (**C**,**D**) from natural enrichment experiments (**A**,**C**) without bubbling and bubble-derived enrichment experiments (**B**,**D**).

### 3.4. Experiment 4—Effect of Different Species of Phytoplankton on Bubbling Enrichment of TEP

This experiment was conducted to study the effect of phytoplankton presence and species-specific differences in TEP enrichment from bubble scavenging (Figure 5). Initial phytoplankton concentrations after dilution into the towers were very low but similar for all species (Table 2). Unfiltered seawater was used for experiment 4, however, similar to previous experiments, TEP, Chl *a* and prokaryote concentrations increased after bubbling in all cases. Nonetheless, while there was no significant difference between the effect of bubbling on the enrichment of Chl *a* between species, the increase of TEP enrichment after 10 min of bubbling was significantly higher for seawater spiked with *T. rotula* (post hoc Tukey, $p = 0.042$, $n = 6$) and *L. polyedra* (post hoc Tukey, $p = 0.013$, $n = 6$), suggesting that additional TEP precursor material was released by *T. rotula* and *L. polyedra* in response to bubbling. TEP concentrations in the SML after 10 min of bubbling increased by $68 \pm 70\%$, $167 \pm 100\%$, $107 \pm 35\%$ and $190 \pm 127\%$ for the control, *T. rotula*, *Phaeocystis* spp. and *L. polyedra*, respectively. TEP concentrations in *T. rotula* and *Phaeocystis* spp. increased after 2 min of bubbling, but did not for *L. polyedra,* even though Chl *a* and prokaryotes did increase for this species after 2 min. Chl *a* concentrations showed no significant difference in enrichment changes from bubbling between the control and species. However, ULW Chl *a* concentrations were significantly higher for all species compared to the control ($p = 0.001$, $n = 3$) which resulted in lower EF, as seen in Figure 5B. Prokaryote concentrations in water spiked with *Phaeocystis* spp. closely matched the control, while *T. rotula* showed greater increases in enrichment after both 2 and 10 min of bubbling (Figure 5C). *L. polyedra* showed lower increases in enrichment after 2 min than what was seen in the control, but higher increases after 10 min. Thus, diatom and dinoflagellate species induced greater increases of prokaryote enrichment in the SML after bubbling.

**Table 2.** Initial phytoplankton cell counts in cultures and resulting diluted concentrations in the towers.

| Species | Culture # | Species concn. in Culture (cell/mL) | Species concn. in Tower (cell/mL) | Species | Culture # | Species concn. in Culture (cell/mL) | Species concn. in Tower (cell/mL) |
|---|---|---|---|---|---|---|---|
| *Lingulodinium polyedra* | 1 | 10,472 | 34 | *Phaeocystis* spp. (free cells) | 1 | 21,201 | 30 |
| | 2 | 8586 | 28 | | 2 | 14,322 | 20 |
| | 3 | 7970 | 26 | | 3 | 23,459 | 34 |
| *Thalassiosira rotula* | 1 | 15,708 | 50 | *Phaeocystis* spp. (colonies) | 1 | 1848 | 3 |
| | 2 | 19,507 | 63 | | 2 | 2772 | 4 |
| | 3 | 16,863 | 54 | | 3 | 2618 | 4 |

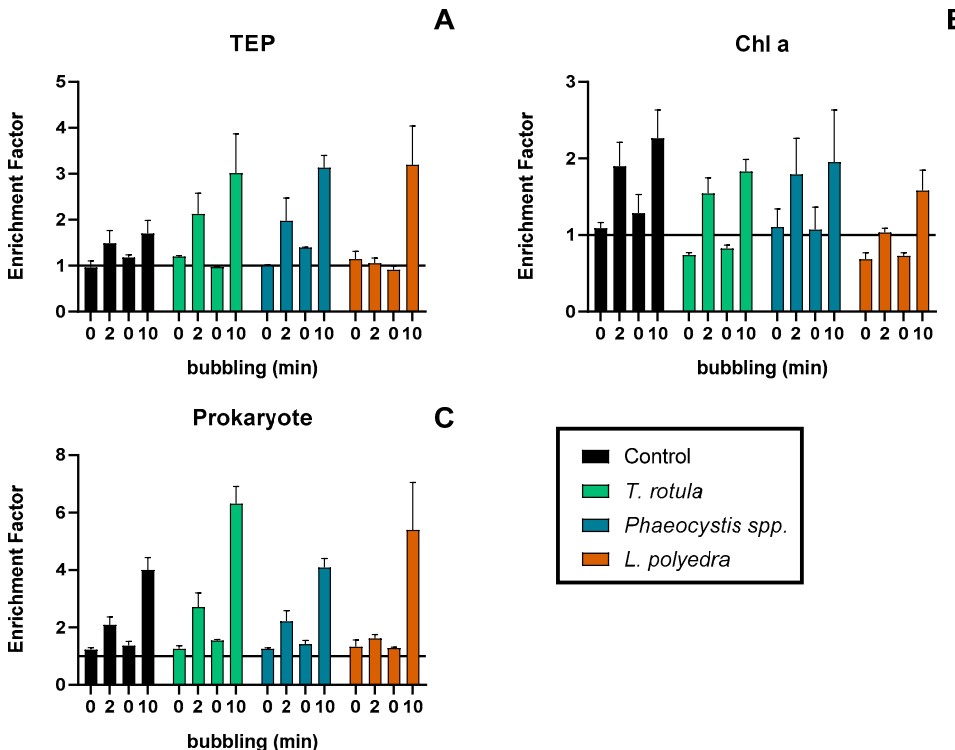

**Figure 5.** Results for the enrichment of TEP (**A**), Chl *a* (**B**) and Prokaryotes (**C**) from experiments using seawater spiked with different phytoplankton species. SML enrichment factors (enriched when EF > 1) from initial sampling (0), after 2 min bubbling (2), initial sampling again (0) and after 10 min bubbling (10). Note differing *y*-axis scales.

## 4. Discussion

TEP is one of the main drivers for OM transfer in the ocean [44] and for the formation of biofilm-like properties of the SML [45,46]. The importance of the SML in air–sea interactions and as a source for sea spray aerosols (SSA) has been extensively supported [1,3,6,7,47]. Furthermore, ever since the SML was shown to maintain enrichment under wave breaking conditions [48] literature has assumed that wave breaking and resulting bubbles must enrich the SML via bubble scavenging mechanisms and increase or alter SSA via bubble bursting mechanisms. However, there are limited studies on the effects that bubbling and bubble bursting have on SSA [25–28] and even fewer studies on the effects of bubble scavenging on the enrichment of the SML [22,23,25]. Therefore, there is a need to better understand the effect breaking waves and bubbling have on SML enrichment, especially for gel particles like TEP, which act as the main transport mechanisms for OM within and between the ocean and atmosphere. This study combines multiple lab experiments with changing physical and biological parameters in order to better understand what bubble characteristics are important for bubbling efficiency on TEP formation and upward transport towards the SML.

### 4.1. How Bubble Size Affects TEP Enrichment

It is assumed, although as of yet unsupported, that bubbling brings TEP, OM and other surface-active material up to the SML by their collection on the surface of bubbles. Another possibility is that this material is entrained, either attached to gel particles or unattached, between bubbles and brought to the SML, where bubbles then become encased by these particles [49,50]. Either way, bubbling has been shown to bring material from ULW to the SML [22,23,25]. We found no significant difference in TEP enrichment in the SML with different bubble sizes, with this having interesting implications. Assuming the process of formation and aggregation of TEP on and between rising bubbles is the

major pathway for TEP enrichment in the SML, if material is entrained on bubble surfaces, then given equal flow rates, the smaller bubbles would show significantly higher increases due to an increased total surface area. For example, during our bubble size experiment, a flow rate of 15.5 L/min was used for all towers, which was based on average bubble sizes measured and a total bubbling time of 2 min, resulting in total surface areas of $3.17 \times 10^8$, $2.62 \times 10^8$ and $1.17 \times 10^8$ for our small (0.58 mm), medium (0.68 mm) and large (1.35 mm) bubbles, respectively. Likewise, if aggregation of material occurs in the space between bubbles because of water suppression and collision, then smaller bubbles should again have higher increases due to an increased number of bubbles. This idea is supported by early studies, which found enrichment of surfactants in the SML and their subsequent release into the atmosphere to be greater with bubbling from small bubbles [51,52]. It is proposed that this is caused by an increased input of film drop to jet drop ratio [53,54]. However, we observed the opposite with a greater enrichment, although not significant, from large bubbles, and no increase of enrichment from small bubbles. Therefore, there must be additional factors masking the effect of bubble size on TEP enrichment. Such factors could be acting against aggregate formation in the SML and/or increasing its release via bubble bursting from small bubbles, or enhancing aggregate formation in the SML and/or suppressing TEP release via bursting from large bubbles. Regardless, we observed no significant influence of bubble size on TEP enrichment in comparison to the large influence that the bubbling rate had.

### 4.2. Effect of Bubbling Rate on TEP Enrichment

We found a large difference between the efficiency of bubbling on enrichment, with low bubbling rates of $8.30 \times 10^7$ bubbles/min or $8.87 \times 10^7$ bubbles/min (14.5 or 15.5 L/min), and high bubbling rates of $9.45 \times 10^7$ bubbles/min (16.5 L/min) ($p < 0.0001$, $n = 15$). The highest bubbling rate caused a doubling of TEP enrichment in the SML within the first 5 min, and increased by six orders of magnitude after 60 min of bubbling and then quickly decreased after the bubbling was stopped. This rate of increase matches a previous study by Robinson et al. [23]. The loss of TEP from the SML after bubbling seemed to be linked to the overall TEP abundance, i.e., experiments with the highest bubbling rate caused the highest TEP abundance in the SML (Figure 2A). Overall, our experiments suggest that the frequency and strength—i.e., bubbling rate—of breaking waves are more important than potential changes in bubble size distributions on the new formation and upward transport of TEP towards the SML.

### 4.3. TEP Enrichment over Time with and without Bubbling

For both experiments with fleece-filtered and unfiltered seawater, we found increasing TEP enrichments with increasing time of bubbling, i.e., after 2 min and 10 min of bubbling. This suggests that TEP formation and transport occured rapidly (<2 min) and increased over time. Furthermore, results from experiment 1 showed that for all bubbling rates, TEP was increasingly enriched over time up to 60 min. Results from Zhou et al. [22] found similar increases of TEP up to 30 min and then observed decreasing TEP due to limited abundance of precursor material left after removal by bubbling. Both studies have shown the abundance of precursor material to be the limiting factor in bubbling efficiency rather than time. Because our study did not produce such high volumes of bubbles to create foam, precursor material was probably not removed as quickly and thus did not result in material constraint after 30 min. A further result from experiment 3 was the enrichment of TEP in the absence of a physical motivator such as bubbling. Results (Figure 4A,B) showed that because of their positive buoyancy, TEP will rise up and become enriched in the SML after approximately 20 min in stagnant water where bubbling is absent. Similar enrichment rates were observed by Azetsu-Scott and Passow [55] using particle-free "clean" TEP. They determined that positive buoyant TEP (density 0.70 to 0.84 g/cm$^3$) moved upward in smaller columns (500 mL volume, 7 cm diameter) with an average velocity of $1.6 \times 10^{-4}$ cm/s and saw upward enrichment after 30 min. Our results show that abiotic formation and enrichment of TEP in the SML can occur without bubbling, even in the presence of low

microbial and biological activity in fleece-filtered source water. However, this enrichment is enhanced by bubble scavenging with an abundance of precursor material as the limiting factor.

### 4.4. Effect of Phytoplankton Presence and Species Difference on TEP Enrichment

Previous studies have found diatoms, dinoflagellates and *Phaeocystis* species to produce TEP in the ocean [11,21,56]. Most recent studies have focused on the production by diatom species [15,57,58], showing the proficiency of this species to produce large amounts of TEP. Therefore, we chose a diatom species (*T. rotula*) [13] and *Phaeocystis* spp. [59] known to produce TEP. In addition, we chose a dinoflagellate species (*L. polyedra*) with an unknown contribution to the TEP cycle but as a common bloom-forming species which is used as a model organism for dinoflagellate physiology and ecology [60]. Similar to one of the first species-dependent studies by Passow et al. [14], our results found that seawater spiked with a dinoflagellate (*L. polyedra*) had comparable enrichment potential to the seawater spiked with a diatom (*T. rotula*). Both yielded greater enrichment from bubbling than the control, and showed the greatest increase after 10 min of bubbling. However, no increase in TEP enrichment was observed for *L. polyedra* after 2 min of bubbling. *L. polyedra* is mobile, while *T. rotula* is a chain forming diatom, thus it could be that *L. polyedra* has some ability to avoid disruption of cells by bubbles and so it takes longer for TEP precursor material to be released. Additionally, they also would have some ability to avoid aggregation on/between bubbles and thus stay in the ULW a little longer than *T. rotula*. We also spiked seawater with a *Phaeocystis* spp., and TEP enrichment was greater than the control for all sampling times, showing a positive effect of this species on TEP enrichment from bubbling. However, Passow and Alldredge [10] reported that intact colonies are not stainable by Alcian blue. Thus, the TEP concentrations from our study would be underestimated, although EF (as a ratio between SML and ULW) is likely to be the same. Comparing the increases of TEP concentration and EF after bubbling between the filtered seawater (experiments 1–3) and unfiltered seawater spiked with phytoplankton (experiment 4), it becomes clear that while the abiotic pathway for TEP formation and/or transfer does utilize bubble scavenging, it is enhanced in the presence of biological activity as a source of precursor material resulting in greater increases of TEP enrichment.

### 4.5. Enrichments of Organic Materials in the SML and Its Air–Sea Transfer

In a recent paper, Marks, et al. [61] suggested that anionic bubble surfaces attract cells with typical negative charges at their outer membranes. It explains the well-known fact that marine bacteria and phytoplankton are concentrated on aerosols created by bubble bursting at the ocean's surface [62]. Bigg, et al. [63] found bacterial cells in aerosols, which were embedded in a gel-like matrix (see image 5 in Bigg et al. [63]). It is consistent with the typical enrichment of TEP in the SML and its biofilm-like features [3]. Indeed, bacteria inhabiting the SML may actively produce TEP as protection from ultraviolet radiation [64]. In addition, the termination of marine phytoplankton blooms provides large amounts of organic matter to the surface layer of the ocean with increasing concentrations in the SML. This organic matter may aggregate to polymeric material, including TEP [48]. Coupling to atmospheric forcing via wind-generated waves carries a fraction of the organic matter via sea-spray generation as aerosols to the atmosphere [65]. It is unknown to what extent transferred TEP remain in the atmosphere or whether they are deposited back to the SML. This probably depends on atmospheric features immediately above the SML, which remains challenging to assess. To our best knowledge, no reports on TEP abundance in aerosols exists, which would further support its transfer from the ocean to the atmosphere. However, further studies are required for a more mechanistic understanding, as rising bubble plumes are an important transport vector in the climate-relevant process of marine aerosol production. Bridging laboratory studies to the field is challenging, but new approaches are emerging [66].

## 5. Conclusions

While the enrichment of TEP in aerosols from bubble bursting has been previously proven, we can now show that the process of bubbling rapidly transports TEP up to the SML within 2 min, and prevents the sinking of TEP aggregates. The rate at which enrichment of TEP and microbial abundance increased from bubbling was found to be dependent on bubbling rates. The availability of precursor material in the case of TEP, rather than bubbling time, was the limiting factor. Literature on wave breaking dynamics has shown there to be a change of bubble size range from breaking waves based on formation, but our results suggest that effects of bubbles size and thus wave formation is less important for TEP enrichment than sustained time of wave breaking and resulting bubble abundance. Lastly, experiments using unfiltered seawater spiked with phytoplankton culture found both a diatom (*T. rotula*) and a dinoflagellate (*L. polyedra*) to have comparable enhanced effectiveness on the enrichment of TEP from bubbling. Thus, not only is the abiotic formation of TEP increased by bubbling, but also by the increase of precursor material available.

**Author Contributions:** Conceptualization, T.-B.R. and O.W.; data curation, T.-B.R. and H.-A.G.; formal analysis, T.-B.R. and H.-A.G.; funding acquisition, O.W.; investigation, T.-B.R.; methodology, T.-B.R.; project administration, O.W.; supervision, O.W.; writing—original draft, T.-B.R.; writing—review and editing, T.-B.R., H.-A.G. and O.W.

**Funding:** This study received funding from the Leibniz-Society (MarParCloud, SAW-2016_TROPOS-2). The APC was funded by the University of Oldenburg.

**Acknowledgments:** We would like to thank Rolf Weinert for the *Phaeocystis* spp. cultures and Christian Spindler for *T. rotula* and *L. polyedra* cultures.

**Conflicts of Interest:** The authors declare that they have no conflict of interest.

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
