# Peer review of "Riding the Plumes: Characterizing Bubble Scavenging Conditions for the Enrichment of the Sea-Surface Microlayer by Transparent Exopolymer Particles"

_atmosphere, doi:10.3390/atmos10080454_

Round 1

Reviewer 1 Report

This manuscript describes a study of parameters that can enrich the concentration of transparent exopolymer particles (TEP) in the sea surface microlayer (SML) in by bubble formation in the underlying water (ULW). While a number of parameters have been considered individually in previous studies, and enrichment of the SML is a general presumption, the significance of this semiquantitative study is asserted to be as systematic examination of how bubbling conditions and certain environmental factors, such as biological sources, contribute to enrichment. The parameters considered are bubble size, concentration, bubbling duration, bubbling rate and in addition, dependence of SMP enrichment on biota species. Three species of phytoplankton representing likely as sources of TEP were tested. The results of this work, though not novel, do move the field forward and thus the report is worthy of publication.

There are a typographical errors, incorrect English usage and confusing statements that are enumerated below and should be addressed by the authors prior to publication.

l. 41: material → materials

l. 119: was → were

l. 119: it is → its

l. 126: (Eq. 1) F*(V/1000) → F*(1000/V)?

l. 133: hinze → Hinze

l. 147 – 148: It is not clear how “optimum” is defined. From context, it appears to mean conditions where maximum transport was observed. If this is the case, then “optimum” is not the appropriated descriptor.

l. 149: “combined with” – do the authors mean “compared with”?

l. 169: This is confusing as written. The reviewer interprets this to mean the towers were filled to 75% of capacity.

l. 197: protocol…were → protocol…was

l. 213: tukey → Tukey

l. 226 invariable → constant

l. 226 ml → mL

Figures 2B, 2C: Can the authors add error bars to these estimates?

l. 262: The heading should be revised to include Chl a.

l. 332 difference on → difference in

l. 333 and following: For smaller bubble size, there are more bubbles for a given flow rate and this would result in greater bubble surface area. Is this the meaning of the statement? If this is so, the authors should specify identical flow rates. Otherwise, some additional explanation is necessary.

l. 340 – 342: The meaning of this statement is unclear with respect to the discussion which immediately precedes.

l. 343: The meaning here is also unclear. The process of aggregation was implied as occurring in the ULW.

l. 355: Fig. 1a → Fig. 2a.

l. 355 – 356: “Overall, our experiments showed that the number of bubbles in a plume, more than the bubble size, plays a key role in the transport and formation of TEP in the SML.” This statement is confusing and should be deleted. The data support the critical factor as the rate of flow. The bubble number is a function of both flow and bubble size.

Reviewer 2 Report

In this article, Robinson and coauthors describe a series of experiments in which a bubble tank with differing bubble sizes, bubbling rate, bubbling duration, and seawater composition was used to determine the impact on transparent exoploymer particle (TEP) enrichment at the sea-surface microlayer.  The results of these experiments where definitive; the enrichment of TEP was enhanced by bubbling and scaled with bubbling rate, bubbling duration, and the presence of certain phytoplankton species.  Although I have few comments on the results of the experiments, I've included several comments below regarding the experimental setup that I'd like to see addressed in a published version.

1) It's not clear whether this setup of constant bubbling is representative of real world conditions or is merely supposed to help the readers understand the mechanisms of bubble scavenging.  If real world conditions are intended, I have the following questions:

    a) Is it realistic to have 2+ minutes of constant bubbling?

    b) Does the higher rate of bubbling represent greater wave action, and if so would the wave         action disrupt the enrichment?

    c) Are any of the bubble sizes tested similar to real world conditions?
    d) Does this setup account for air-sea transfer, and how important is this process for reducing     enrichment?

    e) Is the bubble concentration from any of the bubble rate experiments representative of real     world conditions?

2) Because many of the readers of Atmosphere (including myself) are interested in the air-sea transfer of organic material from the ocean, I'd like to see more discussion of how the enrichment of TEP in the sea surface microlayer increases the amount, particle size, or fraction of organic material in sea spray.

3) Because there are many different kinds of experiments, I'd like to see a summary table added stating whether the experiments had statistically significant levels of enrichment relative to the control or each other.  
